# Secure Multiple-Image Transfer by Hybrid Chaos System: Encryption and Visually Meaningful Images

**Ebrahim Zareimani \* and Reza Parvaz \***

Department of Mathematics, University of Mohaghegh Ardabili, Ardabil 56199-11367, Iran
\* Correspondence: zareimani@uma.ac.ir (E.Z.); rparvaz@uma.ac.ir (R.P.)

**Abstract:** The secure transmission of information is one of the most important topics in the field of information technology. Considering that images contain important visual information, it is crucial to create a safe platform for image transfer. One commonly employed tool to enhance the complexity and randomness in image encryption methods is the chaos system. The logistic and sine maps are utilized in encryption algorithms but these systems have some weaknesses, notably chaotic behavior in a confined area. In this study, to address these weaknesses, a hybrid system based on the Atangana–Baleanu fractional derivative is proposed. The various tests employed to evaluate the behavior of the new system, including the NIST test, histogram analysis, Lyapunov exponent calculation, and bifurcation diagram, demonstrate the efficiency of the proposed system. Furthermore, in comparison to the logistic and sine maps, the proposed hybrid exhibits chaotic behavior over a broader range. This system is utilized to establish a secure environment for the transmission of multiple images within an encryption algorithm, subsequently concealing them within a meaningful image. Various tools employed to assess the security of the proposed algorithm, including histogram analysis, NPCR, UACI, and correlation values, indicate that the proposed hybrid system has application value in encryption.

**Keywords:** encryption; steganography; chaotic system; multiple images; fractional calculation

**MSC:** 34A08; 68P25

## 1. Introduction

In today's digital world, information security is discussed in almost every social network. Security and the fast transfer of information are two major factors in most social networks. Moreover, the safe storage of information is crucial in many fields, such as healthcare, finance, and the legal sector, where sensitive and confidential information is handled regularly. One of the most widely used methods in the discussion of information security is encryption. In addition, encryption methods are used in social networks. The primary factor that influences users' selection of social networks is security. Social networks that prioritize security measures tend to attract more users. By creating a sense of security for social network users, this method ensures their safety and privacy against various types of attack. Many examples can be given of the use of encryption in data transmission; for example, end-to-end encryption is used in WhatsApp. Considering the privacy of patient information, medical images must be protected from access by different parties. The encryption of this type of image during transmission and storage can prevent public access. Furthermore, implementing image encryption algorithms for satellite-captured images ensures a secure transmission process. In addition to these two cases, there are many cases where image encryption has many uses. In recent years, various algorithms have been developed for image encryption; for example, see [1–5]. Various tools have been used for encryption; for example, the chaos system and DNA operations have been used in [6,7], and the variant Hill cipher has been used in [8]. Sometimes, it is necessary to select

a set of images to save and send. This can be seen in medical images taken from different angles to diagnose a disease or in images taken by satellites. Moreover, a 3D image can be taken as a collection of several images. Therefore, creating an encryption algorithm for multiple images would provide a suitable tool for the storage and transfer of images. Various algorithms for the encryption of multiple images have been developed in recent years. In [9], a multiple-image encryption algorithm based on single-channel scrambling is introduced. In [10], following the compression of images via the discrete wavelet transform, the encryption algorithm utilizes bit-level scrambling and diffusion operations on pixels. In addition to the mentioned cases, some algorithms encrypt and compress images to reduce the amount of information sent. However, in these cases, the quality of the sent images decreases, and the original images are not sent. Some algorithms for this method are discussed in papers [11,12]. In addition, video can be considered as a sequence of images, and a chaos system can be used to encrypt them [13,14]. By studying these and similar works, it can be seen that the common feature of the algorithms presented in them is the use of chaotic systems. Therefore, if a chaotic system has a suitable structure, it can improve encryption algorithms. With the development of mathematics in recent years, various encryption tools have been introduced. One of the tools that has attracted researchers' attention in encryption, especially image encryption, is chaos systems. The possibility to create a sequence of numbers that are sensitive to input changes and unpredictability is one of the primary reasons that these systems have garnered the attention of cryptography researchers. Regarding image encryption, this tool has been referenced in several papers, such as [15,16]. Tent, logistic, sine, and cat systems can be mentioned among the most used chaos systems. Among the limitations of these systems are the limited area of chaos and the non-uniform distribution of their outputs. However, algorithms that utilize the one-dimensional chaos system for encryption exhibit certain security vulnerabilities [17,18]. One proposed method to address these issues is to increase the dimensions of the chaotic system [17,18]. In this work, although some problems have been solved, others, such as achieving a uniform distribution, remain unresolved. Moreover, the simplest and, at the same time, the most effective way to solve this problem is to create hybrid systems [19,20]. In this paper, to solve these problems and create a chaos system that has a wider chaos area with a uniform distribution, a hybrid chaos system is introduced using fractional calculations. The use of fractional calculations for the development of chaotic systems has also been employed as a novel solution; for example, see [21–23]. Several tests used to explore the proposed hybrid chaos system demonstrate its efficiency. After introducing and studying the behavior, this system is used in a multi-image encryption algorithm. Considering that an encrypted image has a meaningless structure, an individual who has access to the image can deduce that the sent image is an encrypted image. Subsequently, attackers can attempt various types of attacks on the sent image to decrypt it. Steganography is one of the methods used to hide an image within another image. In this method, an image that conceals another image is called a host image. In recent years, this method has been used to hide encrypted images. First, the image is encrypted, and then the encrypted image is inserted into another image that has meaning. An observer cannot identify the encrypted image concealed within the host image. One method used to hide images is the integer discrete wavelet transform (DWT). In [24], the image is encrypted, and then the encrypted image is hidden in the LH and HH parts of the DWT. A method similar to this has been studied in [25]. In another method, studied in [26], the location of the image pixels is changed by using the Arnold's cat transform and then the image is hidden. One advantage of this method is that if a portion of the image is lost, the adjacent areas in the recycled image are not simultaneously destroyed. In the method described in this article, unlike in the wavelet transform, the host image is scaled to the interval $[0, 1]$, and the numerical values of the pixels are concealed within the decimal digits of the host image's pixels. After encrypting the images using the proposed algorithm based on the hybrid chaos system, the host image is shifted, and then the encrypted images are hidden using decimal representation [26].

The layout of this paper is as follows. In Section 2, using a one-dimensional fractional chaotic system, a novel hybrid chaotic system is introduced. Additionally, in this section, the behavior of these systems is evaluated by various tests, such as the bifurcation diagram and Lyapunov exponent. In Section 3, using the proposed chaos system, a new encryption algorithm for multiple images is studied, and then the encrypted images are changed into visually meaningful images. The simulation results and security analysis of the proposed algorithms are given in Section 4. The conclusions of the paper are presented in Section 5.

## 2. Hybrid Fractional Chaos Generation

In this section, after introducing the preliminary tools, the two-dimensional chaos system is proposed, and, in the next step, the behavior of this system is discussed.

### 2.1. Structure of Proposed Chaos Generation

Chaos systems based on fractional calculations have been studied in various papers. One of the methods used to create a system is the use of the left Atangana–Baleanu fractional derivative, which is studied in paper [22]. In the mentioned paper, a new chaos system is introduced by considering the following fractional equation

$$
{}^{ABR}_{a}D^{\alpha}_{t}x(t) = \Lambda_r(x(t)),
$$

where $\Lambda_r$ represents a hybrid chaos system [27]. Moreover, ${}^{ABR}_{a}D^{\alpha}_{t}$ represents the left Atangana–Baleanu fractional derivative in the Riemann–Liouville sense, which is defined as [28]

$$
{}^{ABR}_{a}D^{\alpha}_{t}f(t) = \frac{\beta(\alpha)}{1-\alpha}\frac{d}{ds}\int_a^t f(s)\varepsilon_{\alpha,1}\left(-\frac{\alpha}{1-\alpha}(t-s)^{\alpha}\right)ds, \quad t \in (a,b),
$$

where $\beta(\alpha)$ and $\varepsilon$ denote normalization and the Mittag–Leffler function, respectively, and are obtained by

$$
\varepsilon_{a,b}(t) := \sum_{i=0}^{\infty}\frac{t^i}{\Gamma(ai+b)}, \quad t \in \mathbb{C},
$$

$$
\beta(\alpha) = 1 - \alpha + \frac{\alpha}{\Gamma(\alpha)}, \quad \beta(0) = \beta(1) = 1,
$$

where $\Gamma$ is a gamma function. In the following, the chaos system that is obtained by this equation is shown by $\varphi_{r,h}^{\alpha}$, where $h$ represents the step length of the partition for $t$. For further details of this type of chaotic system, readers are encouraged to refer to [22].

Now, considering the definitions stated so far, we introduce the hybrid chaos system. Consider $X_0 = (x_0, y_0), r, \alpha$ and $h$ as input values; the proposed system is described in the following steps.

**Step 1.** Obtain $X_0^1 = (x_0^1, y_0^1)$ as

$$
x_0^1 = \varphi_{r,h}^{\frac{\alpha}{2}}(x_0), \quad y_0^1 = \varphi_{2r,h}^{\frac{\alpha}{2}}(y_0).
$$

**Step 2.** Define $r_1$ and $r_2$ as

$$
r_1 = r + \frac{1}{10}\left(\lfloor y_0^1 \times 10 \rfloor - \lfloor y_0^1 \rfloor \times 10\right),
$$

$$
r_2 = 2r + \frac{1}{10}\left(\lfloor x_0^1 \times 10 \rfloor - \lfloor x_0^1 \rfloor \times 10\right),
$$

and then calculate

$$
X_1 = (x_1, y_1) = \left(\mathrm{mod}\left(x_0^1 + \varphi_{r_1,h}^{\frac{\alpha}{10}}(x_0^1), 1\right), \mathrm{mod}\left(y_0^1 + \varphi_{r_2,h}^{\frac{\alpha}{10}}(y_0^1), 1\right)\right).
$$

In the next subsection, the behavior of this system is investigated by various types of tests. Additionally, in the remainder of this paper, the vector of $n$ elements produced by the proposed system is denoted by $\psi_{h,n}^{\alpha,r}$.

*2.2. Behavior Analysis of Proposed System*

In this subsection, as well as in the section related to the simulation of the algorithm, in order to calculate $\varphi_{r,h}^{\alpha}$, the system $\Lambda_r$ is considered as follows:

$$\Lambda_r(x) := \begin{cases} 20 \sin\left(rx(1-x)\right) + \frac{(80-r)x}{2} \quad \mod \ 1, \ \textit{when } x < 0.5, \\[3mm] 20 \exp\left(rx(1-x)\right) + \frac{(80-r)(1-x)}{2} \quad \mod \ 1, \ \textit{when } x \geq 0.5. \end{cases}$$

As mentioned in the Introduction, one of the problems with chaotic systems is the limited chaos area, and the proposed hybrid system addresses this issue. To verify this, consider the 2D logistic map [29] and the 2D sine logistic modulation map (2D SLMM) [18], which are defined as follows:

$$x_{i+1} = r(3y_i + 1)x_i(1 - x_i),$$
$$y_{i+1} = r(3x_{i+1} + 1)y_i(1 - y_i),$$

and

$$x_{i+1} = \alpha\left(\sin(\pi y_i) + \beta\right)x_i(1 - x_i),$$
$$y_{i+1} = \alpha\left(\sin(\pi x_{i+1})\right)y_i(1 - y_i).$$

In the following, these two systems are compared with the proposed hybrid system.

The Lyapunov exponent and bifurcation diagram are the first tests used to investigate the chaotic behavior of the proposed system. The value of the Lyapunov exponent is calculated by $\lambda = \lim_{n \to \infty} \frac{1}{n} \sum_{i=0}^{n-1} \ln |\psi'(x_i)|$ [30]. The relationship between chaotic behavior and this quantity can be expressed as follows [31]: when at least one of the average Lyapunov exponents is positive, the system exhibits chaotic behavior; conversely, if the average Lyapunov exponent is negative, the orbit becomes periodic. A zero-average Lyapunov exponent indicates a bifurcation in the system. The results of the Lyapunov exponent and bifurcation diagram for the 2D logistic map and 2D SLMM and the proposed system for different values of $\alpha$ are shown in Figure 1. By observing these results, it can be seen that the 2D logistic map behaves chaotically from approximately 1.2 onwards, and the 2D SLMM exhibits chaotic behavior from approximately 0.8 onwards. However, the proposed system demonstrates chaotic behavior across the entire range. By comparing the results, it can be seen that the proposed chaos system has a wider chaos area compared to the other two systems.

One of the most important properties for the output of a chaotic system used in an encryption algorithm is that it has a flat distribution. In order to check the distribution of the sequences produced by the 2D logistic map, the 2D SLMM, and the proposed system, the trajectories and histogram plots are drawn, as shown in Figure 2. According to the results of these plots, it is evident that the sequence of numbers produced by the 2D logistic map and 2D SLMM does not exhibit a uniform distribution, nor does it generate a flat histogram. Conversely, the sequence of numbers produced by the proposed system is spread out and exhibits a uniform distribution in the histogram.

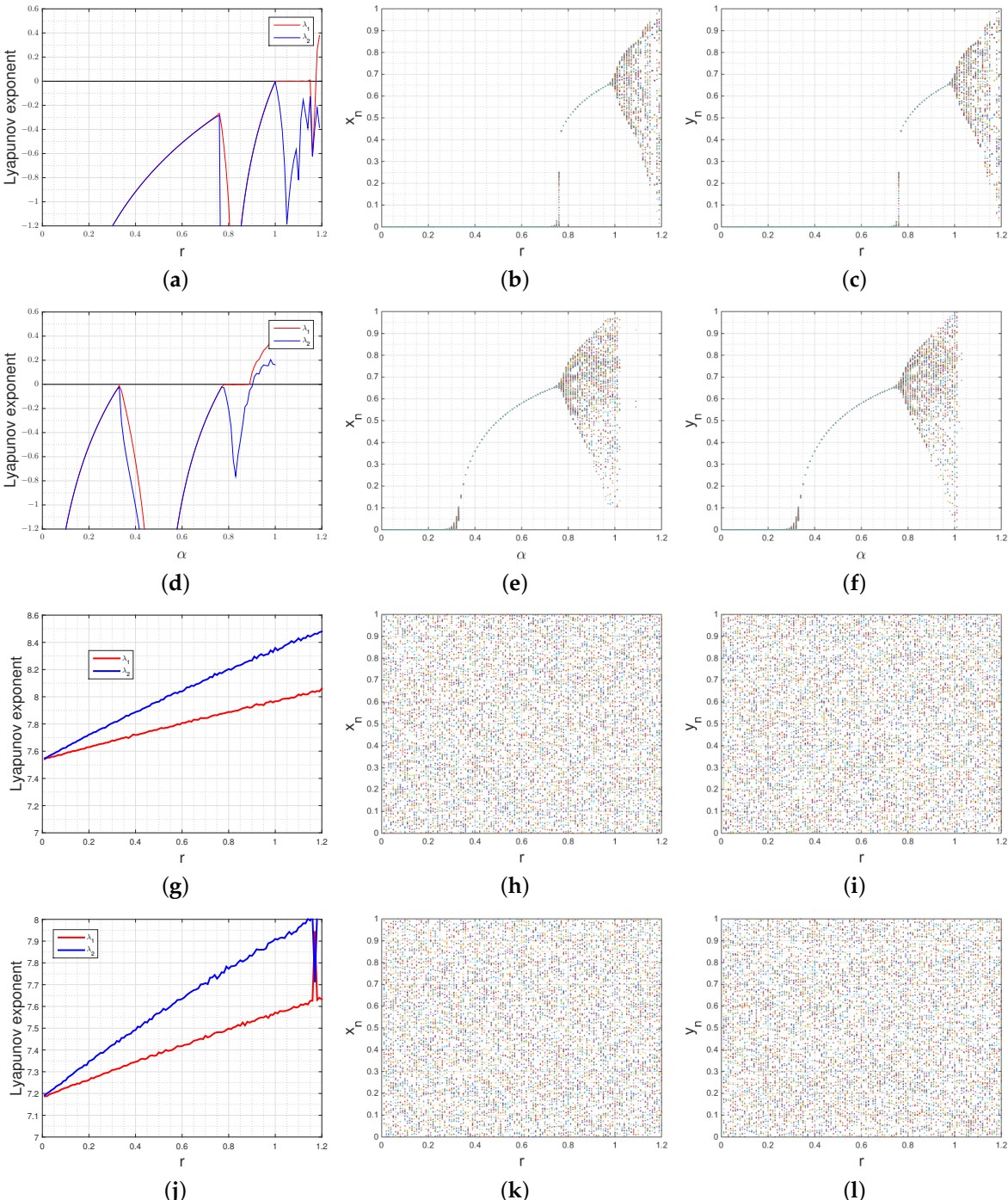

**Figure 1.** Lyapunov exponent and bifurcation diagram: (**a**–**c**) 2D logistic map, (**d**–**f**) 2D SLMM with $\beta = 3$, (**g**–**i**) $\alpha = 0.3$, (**j**–**l**) $\alpha = 0.9$.

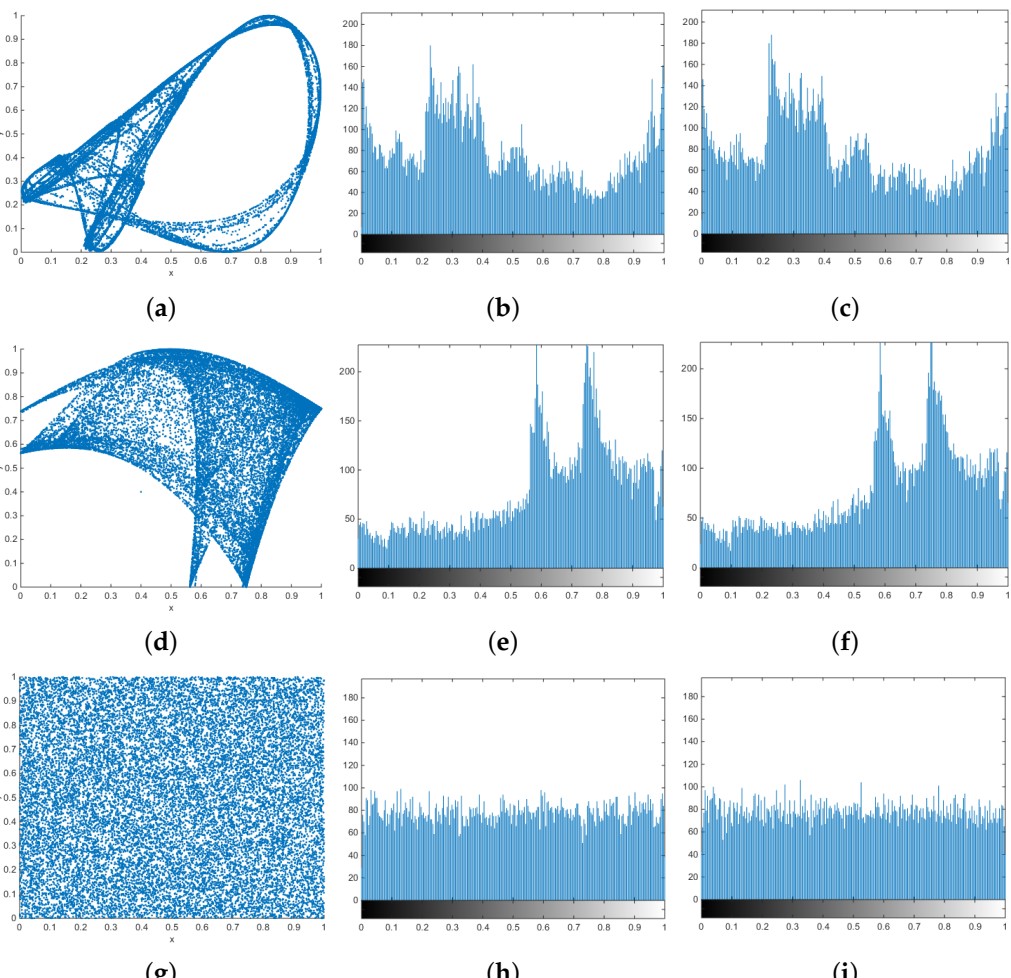

**Figure 2.** Trajectories and histogram plots of (**a**–**c**) 2D logistic map with $r = 1.19$; (**d**–**f**) 2D SLMM with $\alpha = 1$ and $\beta = 3$; and (**g**–**i**) proposed system with $\alpha = 0.3$ and $r = 1.2$.

The sensitivity of the output of the chaos system to the initial value is the most basic feature of the chaos system, providing a suitable tool for use in cryptography. To check the sensitivity to the initial input, the output graphs for different values of $\alpha$ with $r = 1.2$ are drawn, as shown in Figure 3. In this, the output graph is drawn once for the initial values of $(x_1^0, y_1^0) = (0.1, 0.8)$ and again for the initial values of $(x_0, y_0) = (0.1, 0.8 + 10^{-15})$. From Figure 3, it is evident that even a very small change (e.g., $10^{-15}$) in the initial value of the proposed system leads to a completely different generated sequence. Consequently, based on these results, it can be concluded that the proposed system is highly sensitive to the input data.

Now, in the investigation of the behavior of the proposed system, the $0 - 1$ test is studied [32,33]. According to this test, for a time series such as $T_n$, $p$ and $q$ are obtained by using the following system,

$$p_{n+1} = p_n + T_n \cos cn,$$
$$q_{n+1} = q_n + T_n \sin cn, n = 1, 2, \ldots,$$

and $c \in (0, 2\pi)$ is a fixed number. The output of the plots of p versus q for the 2D logistic map and the proposed system are shown in Figure 4. By analyzing the results of this diagram, the chaotic behavior of the system can be studied. Output sequences without chaotic behavior indicate regular dynamics, while output sequences with chaotic behavior indicate irregular dynamics. The 2D logistic map at $r = 1.10$ does not have chaotic behavior

and the plots of p versus q for this map are shown in Figure 4a,b. These results show regular dynamic behavior. This plot is drawn for the proposed system in Figure 4c–f, for different values of $\alpha$. As can be seen, the output of these plots does not have regular dynamics and these results show the chaotic behavior of the proposed system.

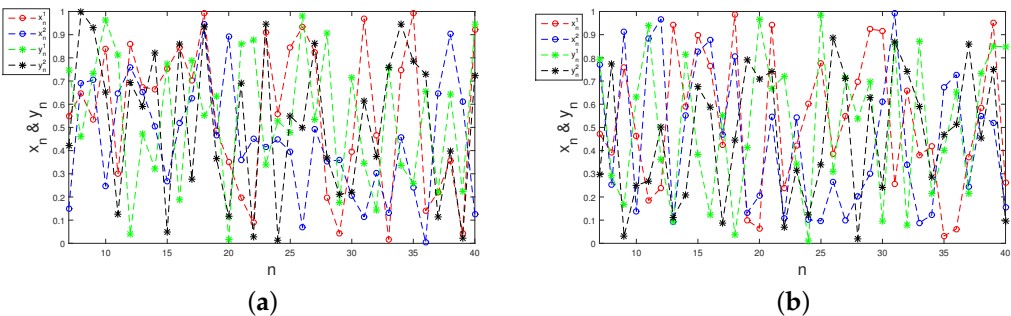

**Figure 3.** Outputs of the proposed hybrid chaotic system with $r = 0.2$ and (**a**) $\alpha = 0.3$, (**b**) $\alpha = 0.9$, and $(x_0, y_0) = (0.1, 0.8)$ (red and green), $(x_0, y_0) = (0.1, 0.8 + 10^{-15})$ (blue and black).

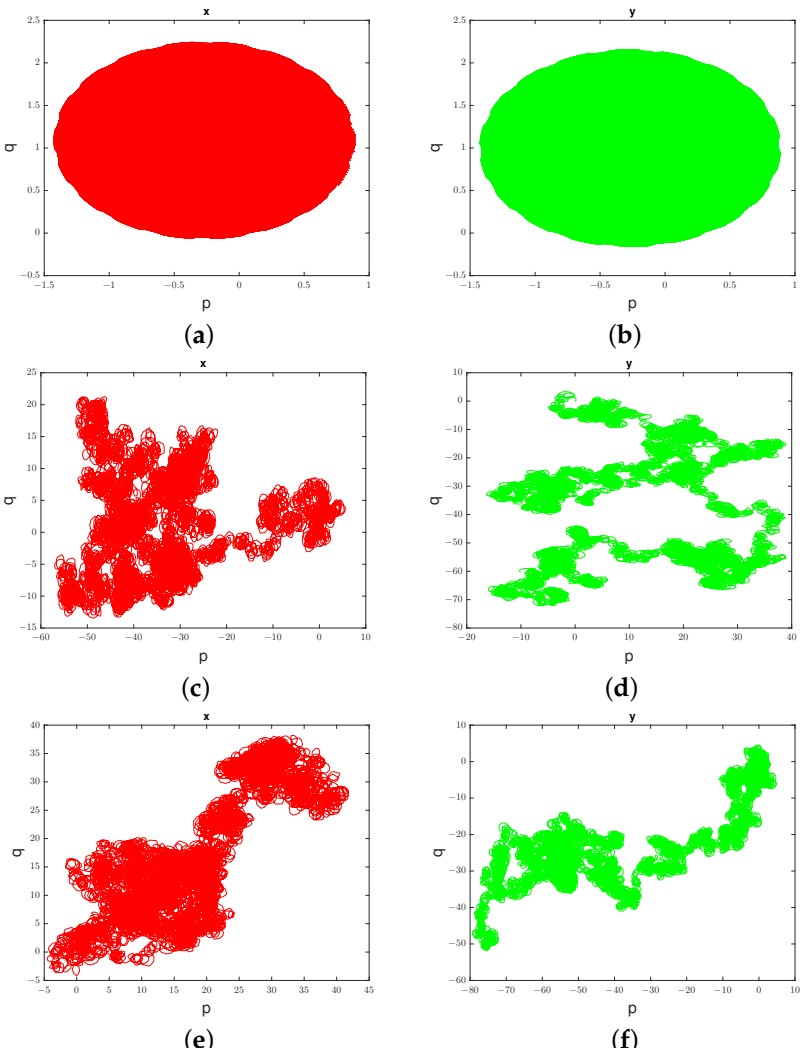

**Figure 4.** Plots of p versus q for the proposed system for (**a**,**b**) 2D logistic map with $r = 1.10$; (**c**,**d**) proposed system with $r = 1.2$ and $\alpha = 0.3$; (**e**,**f**) proposed system with $r = 1.2$ and $\alpha = 0.9$.

One of the most important features of chaotic systems is their ability to generate a sequence of numbers with a random-like structure that can be replicated by specifying the initial values of this sequence. However, how can one ascertain whether a sequence of

numbers truly exhibits the characteristic of randomness? To address this question, various tests exist, such as the TESTU01 or NIST tests. In the following section, we present the results of the NIST test conducted on the sequence generated by the proposed system. This test was introduced by the National Institute of Standards and Technology [34]. The NIST test includes several quantities that are calculated and examines the behavior of the input sequences. For this test, $10^6$ sequences of 100 bits generated by the proposed hybrid system with $\alpha = 0.75, h = 0.001$, and $r = 1.2$ are considered as input. The result for different quantities of this test is shown in Table 1. Based on the results, the evaluated quantities pass these tests with an acceptable percentage.

**Table 1.** The results of the NIST test for $10^6$ sequences of 100 bits generated by the proposed hybrid system.

| | $x_n$ | | $y_n$ | |
|---|---|---|---|---|
| **Type of Test** | ***p*-Value** | **Proportion** | ***p*-Value** | **Proportion** |
| Frequency Test (Monobit) | 0.5264 | 100/100 | 0.4942 | 98/100 |
| Frequency Test within a Block | 0.4848 | 98/100 | 0.4822 | 99/100 |
| Run Test | 0.4662 | 99/100 | 0.5175 | 99/100 |
| Longest Run of Ones in a Block | 0.4933 | 98/100 | 0.4847 | 100/100 |
| Binary Matrix Rank | 0.4467 | 100/100 | 0.4954 | 100/100 |
| Discrete Fourier Transform (Spectral) | 0.4814 | 100/100 | 0.4461 | 100/100 |
| Non-Overlapping Template Matching | 0.5501 | 99/100 | 0.4699 | 99/100 |
| Overlapping Template Matching | 0.4996 | 98/100 | 0.4572 | 98/100 |
| Maurer's Universal Statistical | 0.5474 | 99/100 | 0.4510 | 99/100 |
| Linear Complexity | 0.4596 | 97/100 | 0.4862 | 100/100 |
| Serial Test: Serial 1 | 0.5136 | 99/100 | 0.4795 | 100/100 |
| Serial Test: Serial 2 | 0.4870 | 100/100 | 0.4888 | 100/100 |
| Approximate Entropy Test | 0.5003 | 100/100 | 0.4857 | 99/100 |
| Cumulative Sums (Forward) | 0.5039 | 100/100 | 0.4655 | 98/100 |
| Cumulative Sums (Reverse) | 0.4953 | 100/100 | 0.4710 | 99/100 |

## 3. Proposed Image Encryption Algorithm

In this section, the details of the proposed algorithm are discussed. The proposed algorithm consists of two parts: in the first part, the encryption algorithm is introduced, while, in the second part, the encrypted images are hidden within a meaningful image.

### 3.1. Encryption Algorithm

In this subsection, the details of the proposed encryption algorithm for a sequence of images such as $\{I_i\}_{i=1}^{l} \in \mathbb{R}^{n \times m \times c}$ is introduced. The steps of the proposed algorithm are considered as follows. In the first part, $\{\alpha, h, r, x_0, L\}$, where $L \in \mathbb{R}^{1 \times 4}$, is considered as the initial key, and these keys are perturbed by the following algorithm.

**Step 1.** Consider $\alpha$, $h$, $x_0 \in (0, 1)$, $r \in (0, 2)$, a sequence of images $\{I_i\}_{i=1}^{l} \in \mathbb{R}^{n \times m \times c}$, and $L \in \mathbb{R}^{1 \times 4}$ where each element of the vector $L$ lies within the interval $[0, 1]$ as input values.

**Step 2.** Using the bitxor operator ($\oplus$) in the following stages, the size of the image is reduced to one quarter. At this stage, if the dimensions of the image are not even, they can be made even by repeating the last row or column.

**a.** For $j = 1, \ldots, \frac{n}{2}$ obtain $I_i^1(j, :) = I_i(j, :) \oplus I_i(j + \frac{n}{2}, :)$.

**b.** For $j = 1, \ldots, \frac{m}{2}$ obtain $I_i^2(:, j) = I_i^1(:, j) \oplus I_i^1(:, j + \frac{m}{2})$.

**c.** For $j = 1, \ldots, \frac{n}{4}$ obtain $I_i^3(j, :) = I_i^2(j, :) \oplus I_i^2(j + \frac{n}{4}, :)$.

**d.** For $j = 1, \ldots, \frac{m}{4}$ obtain $I_i^4(:, j) = I_i^3(:, j) \oplus I_i^3(:, j + \frac{m}{4})$.

If the input image is in color, each of these steps will be performed for each image layer. The output sequence of images in this step is shown by $\{I_i^4\}_{i=1}^{l}$.

**Step 3.** Obtain the sequence of matrices $\{A^i\}_{i=1}^{l} \in \mathbb{R}^{20 \times 2}$ by using following steps.

**a.** $A^i(1, 1)$ for $i = 1, \cdots, l$, is calculated as

$$A^i(1,1) = \begin{cases} \mod\left(\frac{\sum_{l_1,l_2} I_i^4(l_1,l_2)}{n \times m}, 1\right), & \text{if } c = 1, \\ \mod\left(\frac{\sum_{l_1,l_2,l_3} I_i^4(l_1,l_2,l_3)}{n \times m}, 1\right), & \text{if } c = 3. \end{cases}$$

**b.** The remaining elements of the matrix sequence are calculated using the following Algorithm 1.

---

**Algorithm 1** Integrating chaos systems into key structures

---

$A^1(1,2) = \mod\left(A^1(1,1) + x_0, 1\right);$
**for** $i = 2 : l$ **do**
$\quad A^i(1,2) = \mod\left(A^i(1,1) + A^{i-1}(1,2), 1\right);$
**end for**
**for** $i = 1 : l$ **do**
$\quad$ **for** $j = 2 : 20$ **do**
$\quad\quad A^i(j, 1:2) = \psi_{h,1}^{\alpha,r}\left(A^i(j-1, 1:2)\right);$
$\quad$ **end for**
**end for**

---

**Step 4.** Calculate the matrix $B \in \mathbb{R}^{20 \times 2}$ as $B = \sum_{i=1}^{l} A^i$.

**Step 5.** Consider

$$X^1 = \text{circshift}\left(B, \lfloor B(1,1) \times 10^2 \rfloor\right),$$

where "circshift" denotes the circular shifting of elements in a matrix. Then, obtain the final vector $X \in \mathbb{R}^{1 \times 4}$ as follows.

$$X = \mod\left(\left(\sum_{l_1=1}^{10} X^1(l_1, :) + L(1, 1:2), \sum_{l_1=11}^{20} X^1(l_1, :) + L(1, 3:4)\right), 1\right).$$

After performing the calculations mentioned above, the final key is used as $\text{Key} = \{\alpha, h, r, x_0, X_n\}$ in the proposed algorithm. One of the most important features of the described algorithm is the sensitivity to very small changes, which is analyzed later. In the second part, the image encryption algorithm is proposed as follows.

**Pixel permutation, disrupting meaning:** In the initial phase of the algorithm, Steps 1 through 7 are introduced to ensure that the pixels of the image are displaced in such a manner that the original image becomes unrecognizable to the observer. By employing this process, if a portion of the image is lost, the resulting recovered image will remain recognizable.

**Step 1.** Consider input images $\{I_i\}_{i=1}^{l} \in \mathbb{R}^{n \times m \times c}$ and the key space as $\{\alpha, h, r, X\}$.

**Step 2.** Define $a_1 := \lfloor \frac{c \times l}{2} \rfloor + 1$ and obtain $\Omega = \psi_{h,a_1}^{\alpha,r}\left(X(1:2)\right)$.

**Step 3.** Convert the sequence of images into a $c$-by-$l$ block matrix as $I^b$, where the $(i,j)$th block is equal to $I_{i,j}^b = I_j(:,:,i)$.

In the next step, the color layers that correspond to the color images are swapped.

**Step 4.** If $c = 1$, go to Step 5; otherwise, obtain two matrices $A, B \in \mathbb{R}^{3 \times K}$ by the following steps.

**a.** Create the vectors $v_A, v_B \in \mathbb{R}^{1 \times 2a_1}$ by

$$v_A = \text{reshape}\left(\lfloor \Omega(1 : a_1, 1:2)^T \times 10^7 \rfloor, [1, 2a_1]\right),$$
$$v_B = \text{sort}(v_A),$$

where $(\cdot)^T$ denotes the transpose notation and "reshape$(E, [k, l])$" reshapes a matrix $E$ into a $k$-by-$l$ matrix. For this purpose, the internal MATLAB function specified with this name can be used. Additionally, "sort$(v)$" is used to sort the members of the vector $v$ from small to large.

**b.** Consider $A, B$ as

$$A = \text{reshape}(v_A(1, 1 : 3l), [3, l]),$$
$$B = \text{reshape}(v_B(1, 1 : 3l), [3, l]).$$

According to the structures of both matrices, they possess identical numerical values for their elements, with only their locations differing.

**c.** Rearrange the positions of each block in the matrix $I^b$ based on the corresponding element's location change in the matrices $A$ and $B$.

**Step 5.** Shift the rows of matrix $I^b$ in the pattern as shown in Figure 5a. In this type of shift, even and odd rows are shifted by $-10\lfloor \frac{c \times n \times l}{\sum \Omega(:) \times 10} \rfloor$ and $\lfloor \frac{c \times n \times l}{\sum \Omega(:) \times 10} \rfloor$ units, respectively. By using this step, the elements of the images are overlapped.

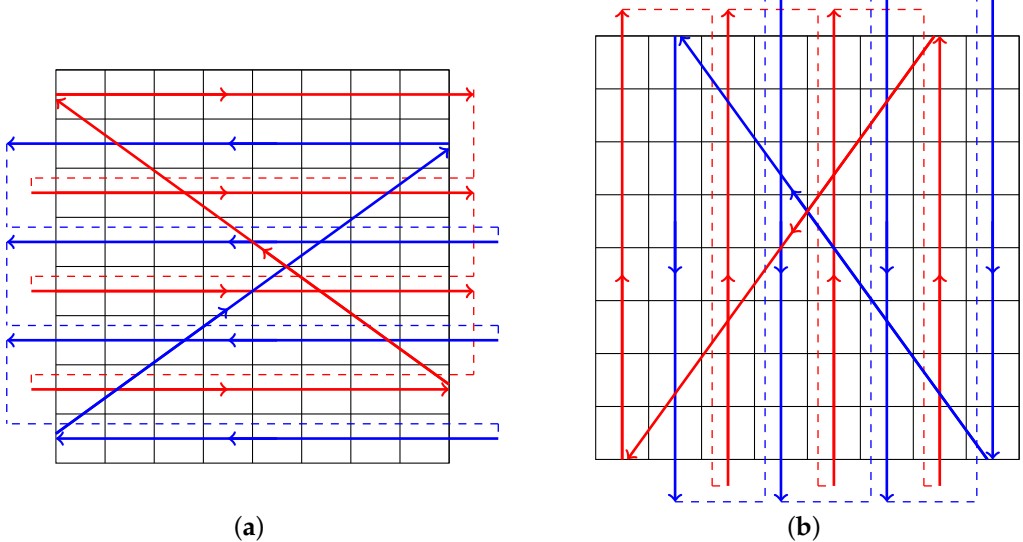

| (a) | (b) |

**Figure 5.** Proposed algorithm shifts: (**a**) Even rows arranged back-to-back and shifted, with odd rows similarly shifted. (**b**) Even columns arranged back-to-back and shifted, with odd columns similarly shifted.

**Step 6.** Shift the elements of matrix $I^b$ by using the following Algorithm 2.

---
**Algorithm 2** Image pixel shift algorithm

---
$a_2 := \max\left(\lfloor \frac{c \times n}{2} \rfloor, \lfloor \frac{l \times m}{2} \rfloor\right);$
$\Lambda = \psi_{h,a_2}^{\alpha,r}(X(3 : 4));$
$s = \text{reshape}(\Lambda^T, [1, 2a_2]);$
**for** $i = 1 : l \times m$ **do**
    $I^b(:, i) = \text{circshift}\left(I^b(:, i), [(-1)^i \times \mod(\lfloor s(i) \times 10^3 \rfloor, c \times n), 1]\right);$
**end for**
**for** $j = 1 : c \times n$ **do**
    $I^b(j, :) = \text{circshift}\left(I^b(j, :), [1, (-1)^j \times \mod(\lfloor s(j) \times 10^3 \rfloor, l \times m)]\right);$
**end for**

---

**Step 7.** Shift the columns of matrix $I^b$ in the pattern as shown in Figure 5b. As in Step 5, in this shift, even and odd columns are shifted by $-10\lfloor \frac{c \times n \times l}{\sum \Omega(:) \times 10} \rfloor$ and $\lfloor \frac{c \times n \times l}{\sum \Omega(:) \times 10} \rfloor$ units, respectively.
**Diffusion:** The impact of these steps (Steps 1–7) on the input images can be observed in the experimental results section. Although the final image resulting from this process successfully obscures the meaning of the original images by displacing their pixels, rendering them

unrecognizable, the histograms of the images remain unchanged. To address this issue, in the following steps, the pixels of the images are combined with sequences generated by the proposed hybrid chaos system.

**Step 8.** Calculate $M_1 = \psi_{h,\lfloor\frac{n}{4}\rfloor\times\lfloor\frac{m}{2}\rfloor}^{2\alpha,2r}\Big((\Omega(a_1,1),\Lambda(a_2,2))\Big)$ and define $M_2 \in \mathbb{R}^{\lfloor\frac{n}{2}\rfloor\times\lfloor\frac{m}{2}\rfloor}$ as

$$M_2\big(1:\lfloor\tfrac{n}{4}\rfloor,1:\lfloor\tfrac{m}{2}\rfloor\big) = \text{reshape}\Big(M_1(:,1),\big[\lfloor\tfrac{n}{4}\rfloor,\lfloor\tfrac{m}{2}\rfloor\big]\Big),$$
$$M_2\big(\lfloor\tfrac{n}{4}\rfloor+1:\lfloor\tfrac{n}{2}\rfloor,1:\lfloor\tfrac{m}{2}\rfloor\big) = \text{reshape}\Big(M_1(:,2),\big[\lfloor\tfrac{n}{4}\rfloor,\lfloor\tfrac{m}{2}\rfloor\big]\Big),$$

and then obtain the sequence of images $\{I_i^d\}_{i=1}^l$ by using the following algorithm. For $i = 1, \ldots, l$, obtain

$$I_i^d\big(1:\lfloor\tfrac{n}{2}\rfloor,1:\lfloor\tfrac{m}{2}\rfloor,:\big) = I_i^b\big(1:\lfloor\tfrac{n}{2}\rfloor,1:\lfloor\tfrac{m}{2}\rfloor,:\big) \oplus M_2,$$
$$I_i^d\big(\lfloor\tfrac{n}{2}\rfloor+1:n,1:\lfloor\tfrac{m}{2}\rfloor,:\big) = I_i^b\big(\lfloor\tfrac{n}{2}\rfloor+1:n,1:\lfloor\tfrac{m}{2}\rfloor,:\big) \oplus M_2,$$
$$I_i^d\big(1:\lfloor\tfrac{n}{2}\rfloor,\lfloor\tfrac{m}{2}\rfloor+1:m,:\big) = I_i^b\big(1:\lfloor\tfrac{n}{2}\rfloor,\lfloor\tfrac{m}{2}\rfloor+1:m,:\big) \oplus M_2,$$
$$I_i^d\big(\lfloor\tfrac{n}{2}\rfloor+1:n,\lfloor\tfrac{m}{2}\rfloor+1:m,:\big) = I_i^b\big(\lfloor\tfrac{n}{2}\rfloor+1:n,\lfloor\tfrac{m}{2}\rfloor+1:m,:\big) \oplus M_2.$$

In this step, it should be noted that if either $n$ or $m$ is an odd number, then we calculate the sequence of $M_1$ such that $M_2$ covers the entire quadrant of the images.

**Step 9.** Calculate $\Theta = \psi_{h,n\times m}^{\alpha,r}\Big((\Omega(a_1,1),\Lambda(a_2,2))\Big)$. Then, define

$$A_1 = \text{reshape}\big(\Theta(:,1),[n,m]\big),$$
$$A_2 = \text{reshape}\big(\Theta(:,2),[n,m]\big).$$

Obtain the sequence of images $\{I_i^e\}_{i=1}^l$ by using the following Algorithm 3.

---

**Algorithm 3** Image diffusion using chaos systems

---

   **for** $i = 1 : l$ **do**
      **if** $c = 1$ **then**
         $I_i^e = I_{1,i}^d \oplus A_1 \oplus A_2$;
      **else if** $c = 3$ **then**
         $I_i^e(:,:,1) = I_{1,i}^d \oplus A_1$;
         $I_i^e(:,:,2) = I_{2,i}^d \oplus A_2$;
         $I_i^e(:,:,3) = I_{3,i}^d \oplus A_1$;
      **end if**
   **end for**

---

In the next steps, the results obtained from Steps 8 and 9 are scrambled using the following algorithms to maximize the confusion regarding the pixel locations.

**Step 10.**

a. Define $s_p$ as

$$s_p = \begin{cases} \frac{\sum_{l_1,l_2} I_i^e(l_1,l_2)}{n\times m\times 256}, & \text{if } c = 1, \\ \frac{\sum_{l_1,l_2,l_3} I_i^e(l_1,l_2,l_3)}{n\times m\times 256}, & \text{if } c = 3. \end{cases}$$

b. Define $a_p := \big|\lfloor s_p \times 10^{15}\rfloor - \lfloor s_p \times 10^{15}\rfloor \times 10^{15}\big|/10^4$. This value is considered as the key that is generated within the algorithm. After calculating these values, taking $r := r + a_p, \alpha : \alpha + a_p, X := X + a_p \times [1,1,1,1]$ and $\{I_i\}_{i=1}^l := \{I_i^e\}_{i=1}^l$, Steps 2 to 7 are repeated, and the final encrypted image is obtained.

Figure 6 provides a summary of the encryption algorithm structure presented in this section. The decryption process is exactly the reverse of the encryption process; therefore, the details of the method are not provided.

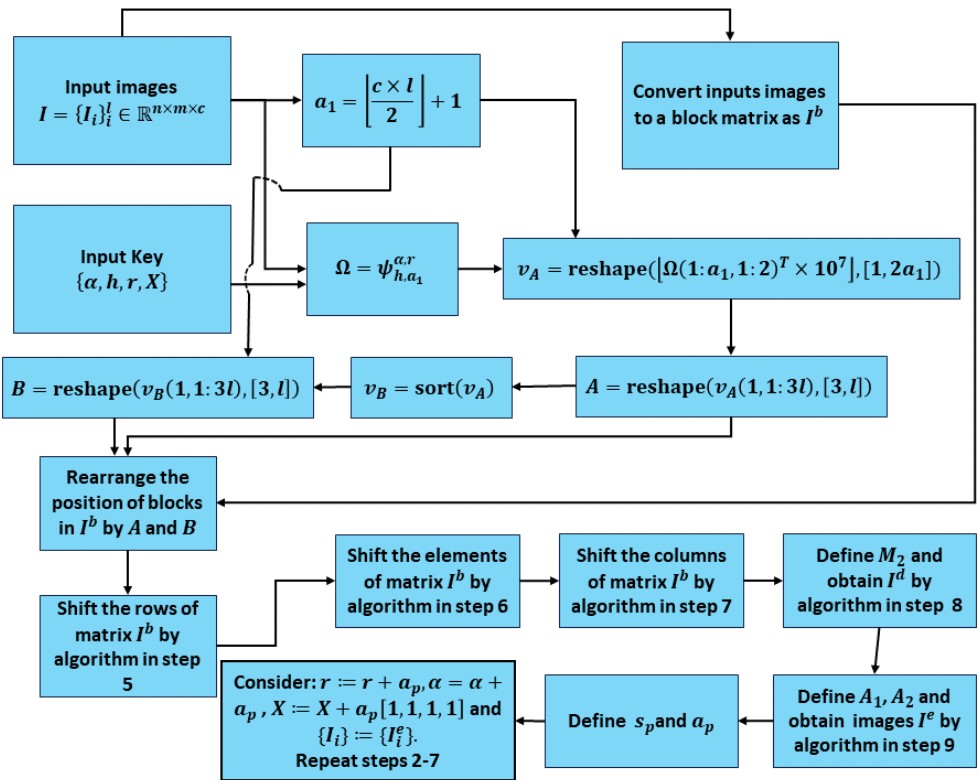

**Figure 6.** Flowchart of the proposed image encryption algorithm.

### 3.2. Visually Meaningful Images

As discussed in the Introduction, there are two common methods of steganography. In the first case, the most significant bits are hidden in the least significant bits of the host image, and the second method involves hiding the main digits among the least significant digits of the host image. In [26], the image values are transferred to the interval $[0, 1]$, and then the digits of the encrypted image that are within the interval $[0, 255]$ are hidden in the least significant decimal digits of the host pixel. In the following, the proposed method is based on this method. The size of the image and the number of places available for hiding are some of the most important challenges in steganography. To explain these challenges, let us assume that the size of the host image is $n \times m$, and the decimal representation of each pixel in the interval $[0, 1]$ is 0. $h_1 h_2 \ldots h_{16}$. Additionally, let us assume that the first four decimal digits, i.e., $h_1, h_2, h_3, h_4$, are considered the most significant digits and should not be changed. Therefore, considering these assumptions, there are $12 \times n \times m$ locations available to hide digits. In this case, four images with a size of $n \times m$, containing pixel values in the range of $[0, 255]$ and integers, can completely fill these places, and, if another image is added to these images, a large image should be selected for the host image. Therefore, if we wish to hide $l$ images with a size $n \times m$, containing pixel values in the range of $[0, 255]$ and integers, the host image must have at least $3 \times l \times n \times m$ places. The number 3 is included in this expression because every integer in the interval $[0, 255]$ is represented by 3 digits. For example, the number 7 is represented as 007 and the number 78 is represented as 078. In the proposed method, if there is only one encrypted image with the size of $n \times m$, the host image is also considered to have a size of $n \times m$, and the pixel values are hidden in the last 3 digits of the decimal representation. However, if the number of encrypted images is up to 4, each with a size of $n \times m$, then the host image is still considered to have the same size. In this case, the last digits of the decimal representation are divided into 3 parts, and the

values of the encrypted images are hidden within these parts. If the number of images increases, the size of the host image will be increased according to the number of images. This enhancement is considered such that each pixel of an image is hidden in only one pixel of the host image. The explanation given up to this point is trivial and is similar to that for the method described in [26]. Now, the proposed steganography method is explained in the following steps.

Consider $H \in \mathbb{R}^{n \times m \times c}$ as the host image, where its pixels are in the range $[0, 1]$. In the following algorithm, $x_{st}, y_{st} \in (0, 1)$ are considered as the keys. Additionally, consider a set of encrypted images $\{I_i^c\}_{i=1}^l$, where the pixels are integers in the range $[0, 255]$. It is also assumed that this host image has enough space to hide the images.

**Step 1.**

**a.** If $c = 1$, consider $H^1 = H$, and if $c = 3$, consider $H^1$ as

$$H^1\big((i-1)n + 1 : in, 1 : m\big) := H\big(1 : n, 1 : m, i\big), \quad i = 1, 2, 3.$$

**b.** Shift the rows of matrix $H^1$ in the pattern, as shown in Figure 5a. In this type of shift, even and odd rows are shifted by $\lfloor x_{st} \times 10^4 \rfloor$ and $-10\lfloor y_{st} \times 10^4 \rfloor$ units, respectively.

**c.** The matrix obtained in the previous part is shifted as in Figure 5b, where even and odd columns are shifted by $\lfloor x_{st} \times 10^4 \rfloor$ and $-10\lfloor y_{st} \times 10^4 \rfloor$ units, respectively.

The output host image in this step is shown as $H^2$.

**Step 2.** At this step, the encrypted images generated by the proposed encryption algorithm are hidden within the host image ($H^2$) using a method similar to [26]. Suppose that the decimal representation of the numerical value of each pixel is $0. r_1 r_2 \ldots r_{15} r_{16}$. In this case, the numerical values of the pixels of the first encrypted image are replaced in $r_{14} r_{15} r_{16}$; the numerical values of the pixels of the second, third, and fourth images are replaced in $r_{11} r_{12} r_{13}$, $r_8 r_9 r_{10}$, and $r_5 r_6 r_7$, respectively. If the number of images increases, a larger host image will be selected, and the images will be hidden using the aforementioned process.

**Step 3.** In this step, reverse Step 1 to obtain the final visually meaningful image.

The general structure of the proposed algorithm is shown in Figure 7.

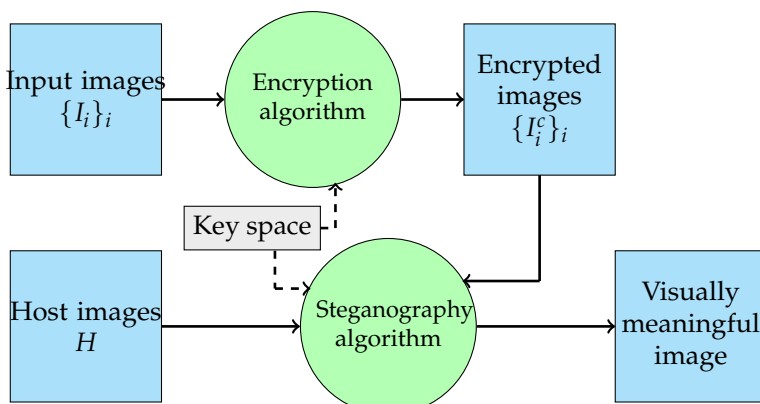

**Figure 7.** Illustration of the proposed algorithm.

## 4. Experimental Results

In this section, the proposed algorithms are simulated and the security is evaluated by different types of tests.

### 4.1. Platform Implementation and Dataset Overview

In the numerical experiment, the proposed algorithm is simulated using Python and the MATLAB 2014b software on Windows 10 64-bit and an Intel(R) Core(TM) i3-5005U CPU @2.00GHz. The USC-SIPI (available online http://sipi.usc.edu/database/ (accessed on 31 December 1977)) image dataset is the source of the images utilized in this section.

Moreover, the code for the encryption part is available on the GitHub site (available online https://github.com/RezaParvaz7 (accessed on 9 April 2024)) of the second author.

### 4.2. Key Sensitivity Analysis

An encryption algorithm should be designed in such a way that the possibility of accessing the key space by attackers is minimized, and the key space should be large enough so that it cannot be compromised by attackers. In addition to these cases, the encryption algorithm should be sensitive to changes in the key space so that it is not possible to recover the original image with very small changes in the key space. In [35], a study was conducted to examine the correlation between the key size and brute force attacks. The findings of this study indicate that if the key space exceeds $2^{100}$, the algorithm is capable of resisting brute force attacks. The proposed key space consists of 9 keys. In addition, two keys are considered according to the visually meaningful image. Thus, the total number of keys in the key space is 11. Considering the precision of the key to be $10^{-15}$ in the calculations, the size of the key space is $10^{165} = 2^{165 \log_2 10} \approx 2^{548}$. This number exceeds the value mentioned in [35]. Therefore, the proposed algorithm is capable of withstanding brute force attacks. In this section, for the numerical results, $I_1$, $I_2$, and $I_3$ are selected as the boat, Lena, and peppers images of size $512 \times 512$, respectively (see Figure 8). In order to assess the sensitivity of the encryption algorithm to the input key, we encrypt images $I_1$, $I_2$, and $I_3$ once with the initial key $key_1$, which is generated using parameters $\alpha = 0.2000$, $h = 0.0010$, $r = 0.4000$, $x_0 = 0.2000$, and $L = [0.5711, 0.5575, 0.7743, 0.6308]$. We then encrypt the same images again with a different initial key, $key_2$, which is generated using parameters $\alpha = 0.2000$, $h = 0.0010$, $r = 0.4000$, $x_0 = 0.2000$, and $L = [0.5711 + 10^{-15}, 0.5575, 0.7743, 0.6308]$. To highlight the differences between the two sets of encrypted images, we calculate the pixel differences for each pixel. The histogram results for these images are presented in Figure 9, which clearly illustrates the disparities between the two sets of images.

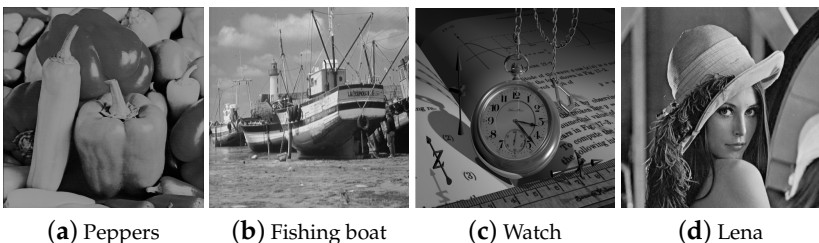

(**a**) Peppers     (**b**) Fishing boat     (**c**) Watch     (**d**) Lena

**Figure 8.** Test images used in the simulation.

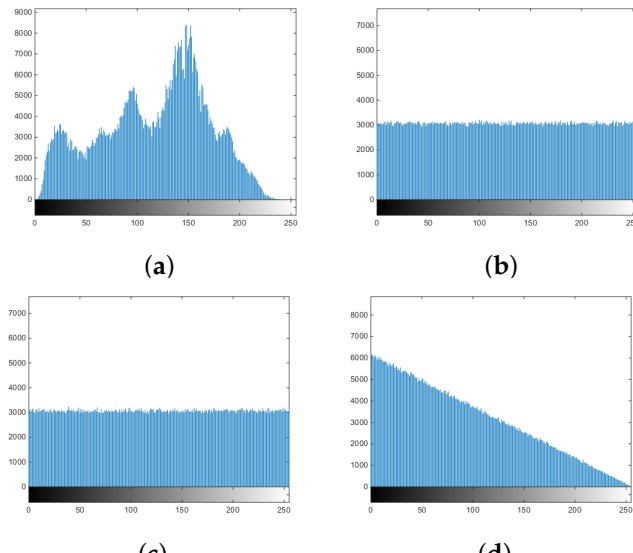

(**a**)

(**b**)

(**c**)

(**d**)

**Figure 9.** Histograms of (**a**) input images, (**b**) images encrypted by $key_1$, (**c**) images encrypted by $key_2$, (**d**) pixel difference.

### 4.3. Statistical Analysis

In this section, various types of statistical tests that are performed on the proposed algorithm are discussed. The distribution of the pixel intensity is visually represented by the histogram. A uniform histogram indicates higher resistance to statistical analysis attacks, making it more challenging for attackers to extract image information. The overall histogram output results for images $I_1$, $I_2$, and $I_3$ are shown in Figure 9, considering two keys. Additionally, to demonstrate the effect of each step of the proposed algorithm on the input images, the output image and the histogram are displayed in Figure 10. Based on the findings, it is evident that the algorithm's output yields a flat distribution. In the results for Figure 10, images 4.1.01, 4.1.02, 4.1.05, and 4.1.06 ($256 \times 256$) are used. For this example, $L = [0.9000, 0.4000, 0.0010, 0.3000]$, $\alpha = 0.9000$, $r = 0.4000$, $x_0 = 0.2$, and $h = 0.001$ are used. In addition, to generate a meaningful image, image 4.1.03 with $x_{st} = 0.2491$ and $y_{st} = 0.2494$ is utilized as the host image and key, respectively. In this figure, in order to better display the sequences of the output images, they are placed together, and the overall histogram of the images is shown. The histograms of the host and visually meaningful images are depicted in Figure 11. Based on the results shown in these figures, it is evident that the encrypted images exhibit a uniform distribution, and there are no significant differences observed in the histograms of the host image and the visually meaningful image.

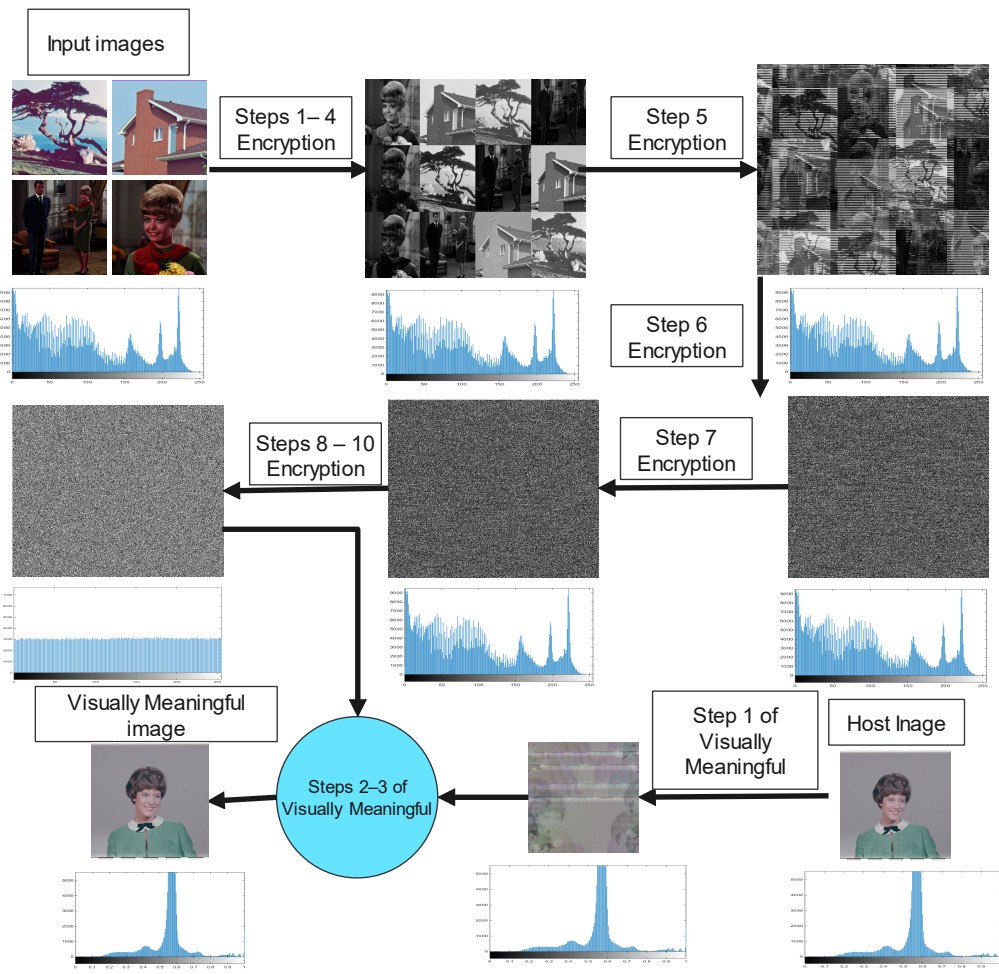

**Figure 10.** The effect of the steps in the proposed image encryption algorithm.

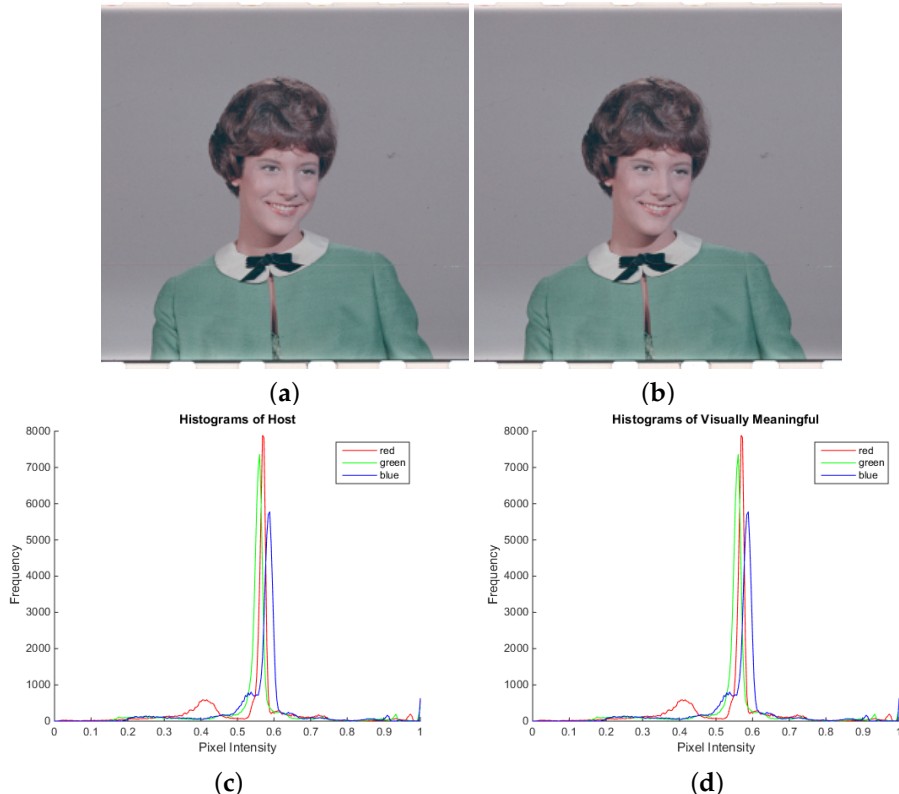

**Figure 11.** (**a**) Host image, (**b**) visually meaningful image, (**c**) histogram of host image, (**d**) histogram of visually meaningful image.

Another test that is used to check the encryption algorithm is the correlation test. Let $E[\cdot]$ symbolize the expectation value, $\mu$ denote the mean value, and $\sigma$ represent the standard deviation; then, this value is obtained by

$$C_{x,y} = \frac{E\left[(x - \mu_x)(y - \mu_y)\right]}{\sigma_x \sigma_y}.$$

The results of these tests are given in Table 2 and compared with the results of other works. Moreover, the correlation distributions for the original images and encrypted images are shown in Figure 12. The correlation distributions for host image 4.1.03 before and after hiding the four images 4.1.01, 4.1.02, 4.1.05, and 4.1.06 are presented in Figures 13 and 14, respectively. These results indicate that there is no visible difference in the outputs. According to these results, it can be seen that the proposed algorithm provides an acceptable representation for this test.

**Table 2.** Correlation coefficient analysis.

| Image | Horizontal | Vertical | Diagonal |
|---|---|---|---|
| Boat | 0.9450 | 0.9758 | 0.9283 |
| Peppers | 0.9732 | 0.9847 | 0.9550 |
| Lena | 0.9666 | 0.9823 | 0.9566 |
| Proposed method | 0.0005 | 0.0001 | 0.0003 |
| Method in [36] | 0.0020 | −0.0006 | −0.0062 |
| Method in [37] | 0.0032 | −0.0182 | −0.0021 |
| Method in [38] | 0.0635 | 0.1981 | 0.1698 |
| Method in [39] | 0.0041 | 0.0043 | 0.0084 |

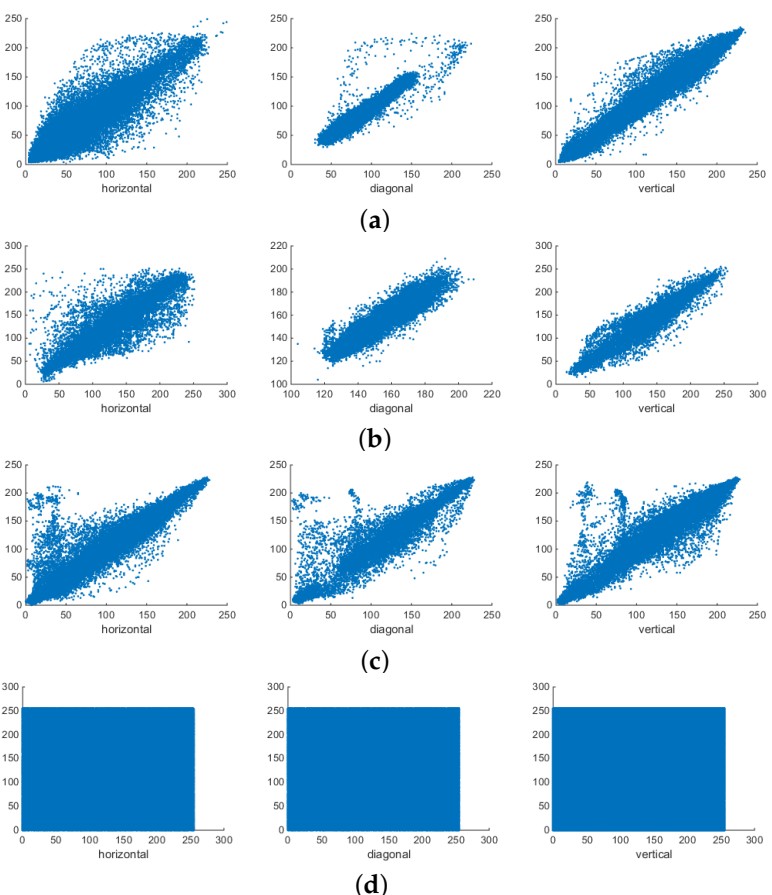

**Figure 12.** Correlation of neighborhood pixels in different directions: (**a**) Lena image, (**b**) boat image, (**c**) peppers image, (**d**) encrypted images.

Among the various quantities that are studied to check an encryption algorithm, there are some values that have an ideal value. The algorithm's representation improves as the values obtained from the studied algorithm approach this ideal value. In the following, these values are studied. By considering $w$ and $P$ as the gray level and the probability, respectively, this value is obtained by

$$H(k) = -\sum_{i=0}^{w-1} P(k_i) \log_2 P(k_i).$$

The output value for this quantity is between 0 and 8, and the ideal value is equal to 8. The results of this test are given in Table 3 and compared with those of the other methods. The entropy of the cipher image exceeds 7.9998 and approaches 8. These results show that the output of the proposed algorithm is close to the ideal value.

**Table 3.** Information entropy analysis.

| Plain Image | Boat | Peppers | Lena |
|---|---|---|---|
| | 7.1914 | 7.4451 | 7.5937 |
| Encrypted image | Proposed method 7.9998 | Method in [36] 7.9998 | Method in [40] 7.9994 |
| | Method in [41] 7.9996 | Method in [42] 7.9995 | Method in [43] 7.9994 |

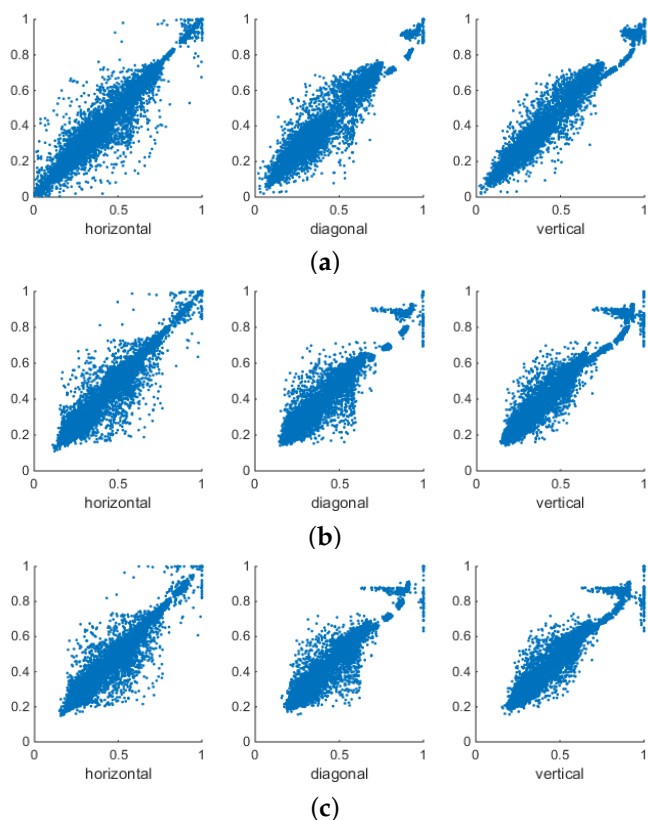

**Figure 13.** Correlation of neighborhood pixels for host image 4.1.03 in different directions: (**a**) red channel, (**b**) green channel, (**c**) blue channel.

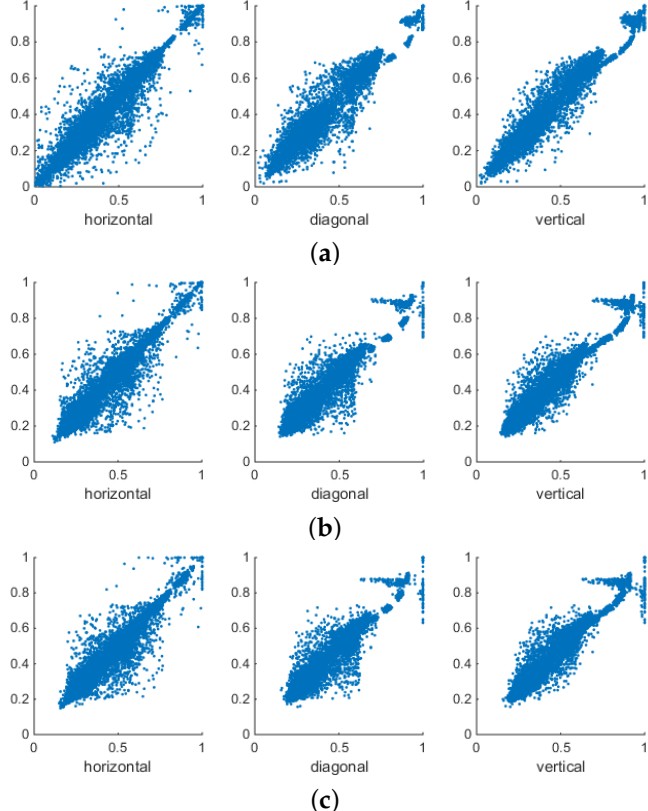

**Figure 14.** Correlation of neighborhood pixels for visually meaningful image 4.1.03 in different directions: (**a**) red channel, (**b**) green channel, (**c**) blue channel.

Sensitivity analysis relies on two crucial tools, namely NPCR and UACI. NPCR, which refers to the number of pixels change rate, quantifies the rate at which pixels change when a single pixel in a plain image is altered. On the other hand, UACI, or the unified average changing intensity, measures the average intensity difference between a plain image and its encrypted counterpart. The ideal values for NPCR and UACI are set at 100% and 33.33%, respectively. When regarding $C^1$ and $C^2$ as matrices of size $n \times m$ representing the encrypted image before and after the modification of the original image, these quantities are defined as

$$NPCR = \frac{\sum_{i,j} D_{i,j}}{m \times n} \times 100\%,$$

$$UACI = \frac{\sum_{i,j} |c_{i,j}^1 - c_{i,j}^2|}{255 \times n \times m} \times 100\%$$

where

$$D_{i,j} = \begin{cases} 1 & \text{if } c_{i,j}^1 \neq c_{i,j}^2, \\ 0 & \text{if } c_{i,j}^1 = c_{i,j}^2. \end{cases}$$

The results related to this test for color and gray images are given in Tables 4 and 5. In these tables, in each row, the name of the image and the corresponding pixel that has been changed are specified. Based on these results, it can be observed that the output values are close to the ideal values.

**Table 4.** NPCR and UACI analysis of grayscale images.

| Image | NPCR | UACI | Image | NPCR | UACI |
|---|---|---|---|---|---|
| Boat (1,1) = 0 | 99.6114 | 33.4516 | Boat (200,200) = 0 | 99.6151 | 33.4622 |
| Peppers (1,1) = 0 | 99.6140 | 33.4915 | Peppers (200,200) = 0 | 99.5978 | 33.4319 |
| Lena (1,1) = 0 | 99.6058 | 33.4842 | Lena (200,200) = 0 | 99.6071 | 33.4679 |

**Table 5.** NPCR and UACI analysis of color images.

| Image | NPCR | UACI | Image | NPCR | UACI |
|---|---|---|---|---|---|
| 4.1.01 (1,1) = 0 | 99.6141 | 33.5025 | 4.1.01 (200,200) = 0 | 99.6150 | 33.4492 |
| 4.1.02 (1,1) = 0 | 99.6108 | 33.4912 | 4.1.02 (200,200) = 0 | 99.6026 | 33.4843 |
| 4.1.05 (1,1) = 0 | 99.6081 | 33.4551 | 4.1.05 (200,200) = 0 | 99.6058 | 33.4911 |
| 4.1.06 (1,1) = 0 | 99.6078 | 33.4418 | 4.1.06 (200,200) = 0 | 99.6049 | 33.4559 |

Encrypted images should have the maximum difference from the plain images. To check the similarity of two images, various tests are available, among which we can refer to the peak signal-to-noise ratio (PSNR) and structural similarity index (SSIM) tests. If the values of the SSIM are close to zero and the PSNR is less than 10 dB, then it can be said that these images are different. In this case, the sequences of the original images and encrypted images are combined, and the results are reported. In this test, the images of the peppers, boat, and watch, with dimensions of $512 \times 512$, are used, and the key space is considered as $\alpha = 0.90$, $r = 0.4$, $h = 0.001$, $x_0 = 0.25$, and $L = [0.5711, 0.5575, 0.7743, 0.6308]$. The results for these tests are presented in Table 6 and compared with the results in [44]. According to these results, it can be observed that the differences between the encrypted images and the plain images are evident. For this example, the sequences of the input, encrypted, and decrypted images, as well as their histograms, are depicted in Figure 15. According to the results, it can be seen that the histograms of the plain and decrypted images are similar, but the histogram of the encrypted image has a uniform distribution. Moreover, the NPCR and UACI values and correlation coefficients are reported and compared in Tables 7 and 8, respectively.

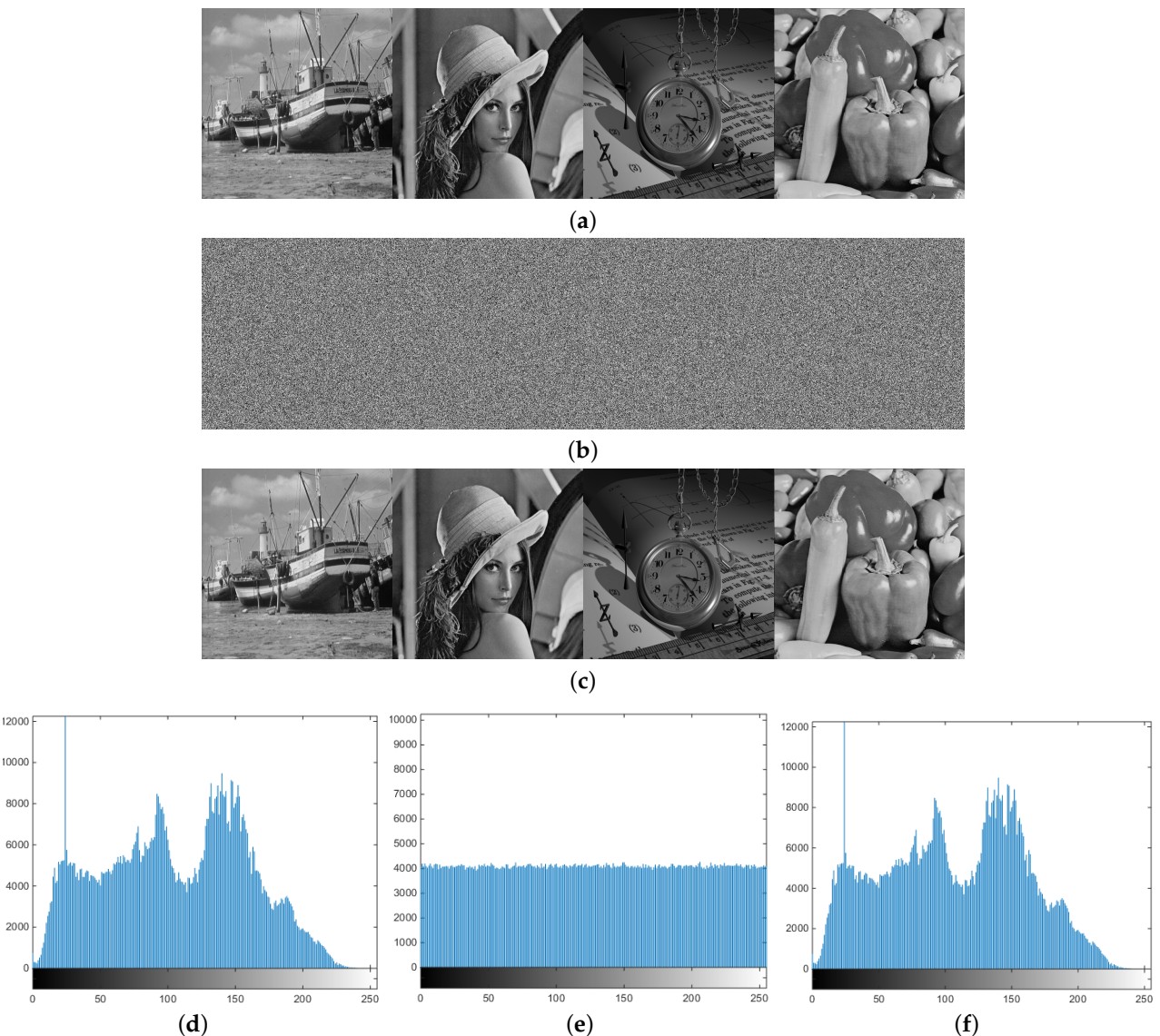

**Figure 15.** Encryption and decryption results: (**a**) input sequence images, (**b**) encrypted sequence images, (**c**) decrypted sequence images, (**d**) histogram of plain images, (**e**) histogram of encrypted images, (**f**) histogram of decrypted images.

**Table 6.** Analysis of encryption quality.

| Method | PSNR | SSIM |
|---|---|---|
| Proposed algorithm | 8.6929 | $2.6971 \times 10^{-4}$ |
| Algorithm in [44] | 8.8260 | $1.3449 \times 10^{-6}$ |

**Table 7.** NPCR and UACI values.

| Method | NPCR | UACI |
|---|---|---|
| Proposed algorithm | 99.6161 | 33.4514 |
| Algorithm in [44] | 99.6060 | 33.5126 |
| Algorithm in [45] | 99.9100 | 33.4800 |
| Algorithm in [40] | 99.6250 | 33.4510 |
| Algorithm in [41] | 99.1841 | 33.5284 |

**Table 8.** Correlation coefficients of encrypted images.

| Method | Horizontal | Vertical | Diagonal |
|---|---|---|---|
| Proposed algorithm | −0.0001 | 0.0012 | 0.0005 |
| Algorithm in [44] | −0.0003 | 0.0011 | 0.0013 |
| Algorithm in [45] | −0.0036 | 0.0026 | 0.0012 |
| Algorithm in [40] | −0.0016 | 0.0057 | −0.0189 |
| Algorithm in [41] | 0.0034 | 0.0015 | 0.0008 |

### 4.4. Noise and Data Loss Attacks

Some attacks are not aimed at decrypting information, but rather at causing destruction, such as noise and data loss attacks. These attacks result in the loss of certain parts of the information. The results for this test are given in Figures 16 and 17. In Figure 16, for the noise attack, salt and pepper noise with a density of 0.2 is used. Additionally, as seen in Figure 17, after the encrypted images are hidden in the host image, a part of the visually meaningful image is lost. Subsequently, the algorithm's extraction steps are executed. Based on the obtained outcomes, it is evident that the encrypted images remain identifiable. Hence, it can be concluded that the suggested algorithm effectively withstands attacks involving noise and data loss.

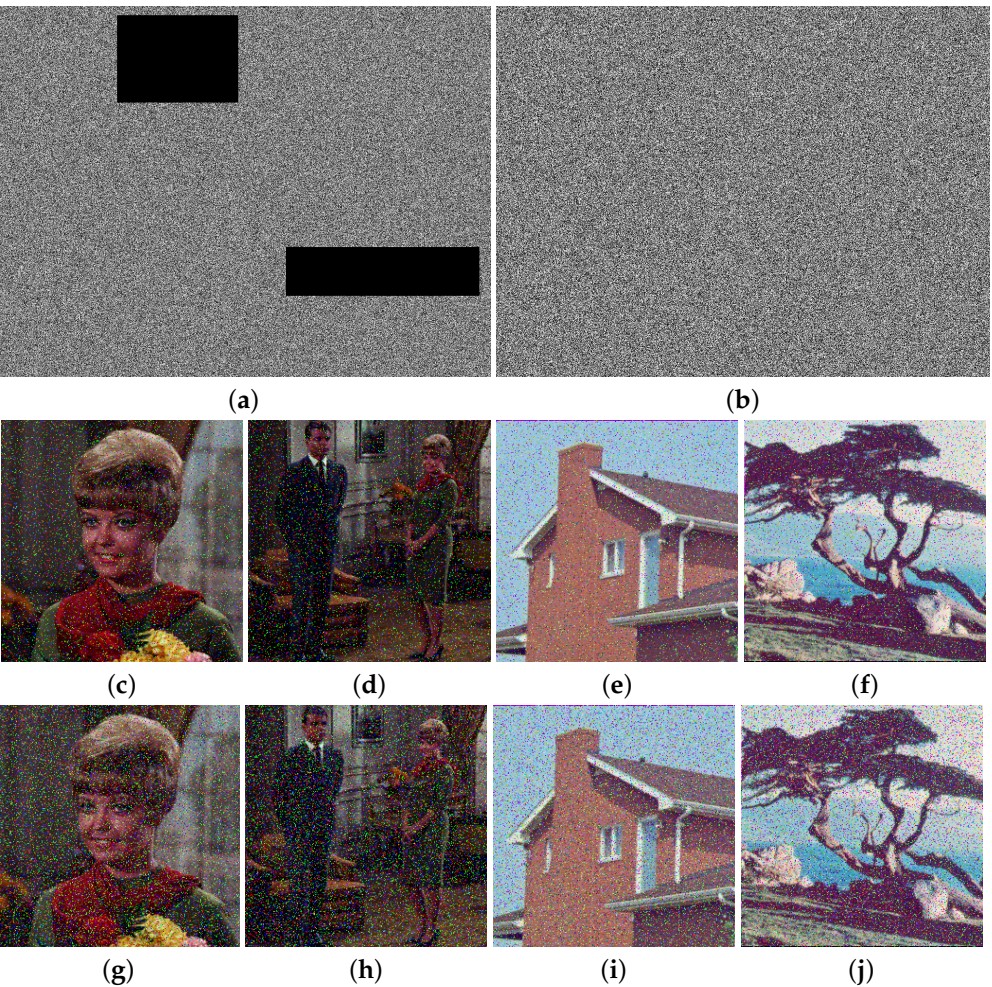

**Figure 16.** Results of the data loss and noise attacks: (**a**) data loss attacked image, (**c**–**f**) results of the proposed algorithm after data loss attack, (**b**) noise attacked image, (**g**–**j**) results of the proposed algorithm after noise attack.

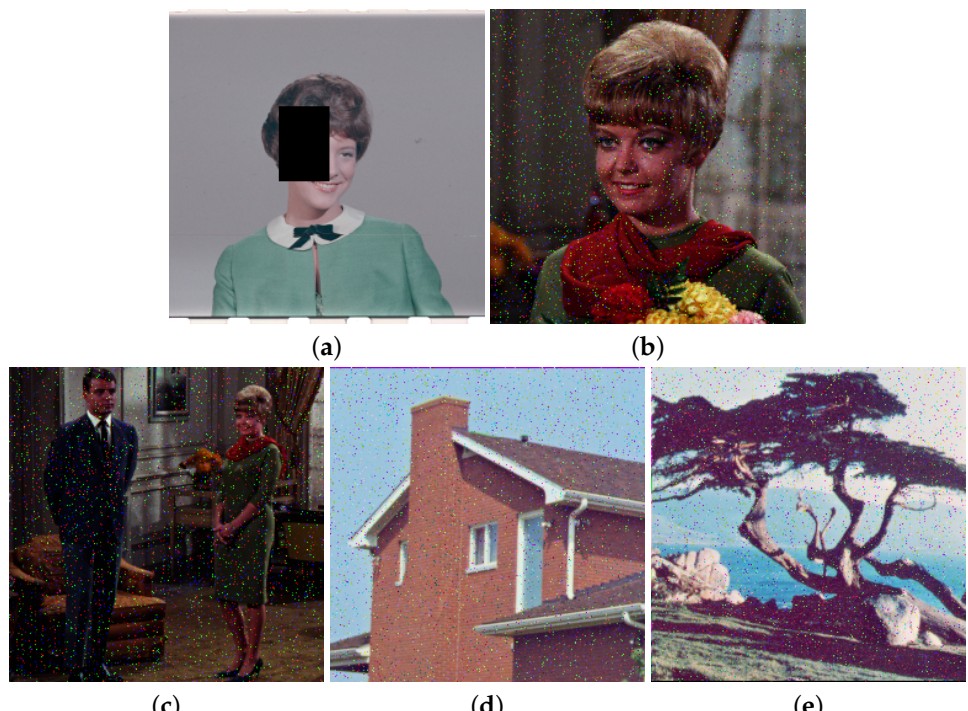

**Figure 17.** Results of the data loss attack for visually meaningful image: (**a**) visually meaningful image after data loss attack, (**b**–**e**) restored images after data loss attack.

## 5. Conclusions

This paper introduces a hybrid chaos system that addresses several issues encountered in previous systems, such as the limited chaos area and the need for a uniform distribution in production sequences. To check the chaotic behavior of the proposed system, various types of tests, including the histogram, trajectory, Lyapunov exponent, and 0–1 tests, are evaluated. The NIST test is also used to check the randomness of the production sequence of the proposed system. The results of these tests show the effectiveness of the proposed system in producing a sequence with a chaotic structure. As an illustration of the application of this system, an algorithm for the encryption and concealment of images within meaningful images is provided. Various statistical and security parameters are utilized to assess the security of the presented algorithm, and the results indicate the efficiency of the proposed hybrid system and the proposed algorithm in the secure transfer of images.

**Author Contributions:** Methodology, E.Z. and R.P.; software, E.Z. and R.P.; formal analysis, R.P.; investigation, E.Z.; writing—review & editing, R.P. All authors have read and agreed to the published version of the manuscript.

**Funding:** This research received no external funding.

**Data Availability Statement:** No new data were created or analyzed in this study. Data sharing is not applicable to this article.

**Acknowledgments:** The authors express their gratitude to the editor and reviewers for their helpful comments.

**Conflicts of Interest:** The authors declare no conflict of interest.

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
