# Peer review of "Secure Multiple-Image Transfer by Hybrid Chaos System: Encryption and Visually Meaningful Images"

_mathematics, doi:10.3390/math12081176_

Round 1
Reviewer 1 Report
Comments and Suggestions for Authors
The paper proposes a hybrid chaos system based on fractional calculations for the encryption and visually meaningful representation of multiple images. After my careful review, I found that there are many problems with the paper, specifically the following problems:
The abstract may be vague or unclear and fail to convey the key information of the study. It is recommended that you revisit the abstract and ensure that you use concise and clear language so that readers can quickly understand the purpose, methods and key findings of the study. In addition, while providing key information, the abstract should also highlight the originality and importance of the research, so that readers can immediately realize the contribution of your research to the field. Please ensure that the abstract contains sufficient background information, a clear statement of the research question, and a concise summary of your research methods and main results. Finally, pay attention to grammar and sentence structure to ensure your abstract is grammatically correct to improve readability. When rewriting, use more direct and specific expressions to ensure that readers are interested in your research.
When reviewing your paper, I found that the contribution statement is indeed missing and fails to clearly highlight the uniqueness and innovation of your research in this field. It is recommended that you express a statement of contribution that clearly points out the novelty of your research relative to the existing literature and its specific contribution to practical applications or theoretical development in related fields. When writing, emphasize what knowledge gaps your research fills, what unanswered questions it solves, and highlight the significance of your methods or findings for the further development of the field. Additionally, provide specific illustrations or examples to support your claim about the innovativeness of your research. Make sure your statement of contribution piques your readers' interest and creates a strong curiosity about your research. The ultimate goal is to make the contribution statement clear and engaging so that readers can accurately understand the value of your research on both an academic and practical level.
The test of the chaotic system proposed in the paper is insufficient. Please supplement the following experiments:
(1) NIST test
(2) TESTU01
Some references in the paper are out of date. Supplementing and discussing the literature in the past three years will help improve the quality of the paper.
I have some concerns regarding this paper's submission. Firstly, the authors appear to not fully understand the fundamental issue of key generation in cryptography, as it is not feasible to generate a secret key from plaintext. In cryptography, the main rule is to have an independent secret key from the plaintext. There should be no relationship between them. It is concerning that a paper in Signal Processing would have the same issue. We need to stop promoting this flawed approach in research papers. I strongly advise you to redesign your algorithm so that there is no relationship between the key and plaintext. I recommend referring to papers that have provided good security results without any relationship between the secret key and plaintext.
While reading your paper, I noticed something unreasonable about the encryption structure. A complete chaotic image encryption system should usually contain key elements such as displacement, diffusion and confusion to ensure effective protection of images. However, your encryption structure appears to be missing processes, which could lead to potential risks in terms of security and reliability. First, the permutation process rearranges the pixels in the image, increasing the complexity and randomness of the encryption, making it more difficult to crack. It is recommended that you introduce a permutation operation into the encryption structure to ensure that it effectively obfuscates the pixels without losing information. Secondly, diffusion ensures that the key and image information are fully mixed through a series of complex transformations, thereby enhancing the strength of the encryption system. Consider incorporating a diffusion step into your structure to better spread the influence of the key, thereby increasing the randomness and security of your encryption. Finally, the obfuscation process often involves introducing chaotic elements into the image, making it more difficult to crack. Make sure your encryption structure contains sufficiently complex obfuscation steps to effectively hide the statistical properties of the image and improve the security of your system. By introducing key elements such as displacement, diffusion, and obfuscation, you can improve the security of your chaotic image encryption system, making it more robust and reliable. Hopefully these suggestions help improve your encryption structure.
The paper lacks a comparison with existing image encryption algorithms, which would provide a better understanding of the novelty and advantages of the proposed algorithm. How does the proposed algorithm compare to other state-of-the-art image encryption algorithms in terms of security and efficiency? The experiments should be supplemented with: (1) PSNR (2) SSIM, etc. You can refer to the experiments of excellent articles.
There should be a clear overall block diagram including the steps of the algorithm. Can be drawn using software such as Visio.
There should be a description of the decryption process. It would be better to give more detailed information based on the encryption process.
Histogram analysis should have encryption and decryption analysis.
and so on.
I am very grateful to the editor for allowing me to review this paper, and I have also seen the contributions made by more colleagues in image encryption. Although this paper has fatal problems, I still gave suggestions very seriously. I hope that the author can gain greater gains after revising it.
Comments on the Quality of English LanguageRequires major modifications.
Reviewer 2 Report
Comments and Suggestions for Authors
The authors propose an encryption algorithm for multiple images using a hybrid chaos system based on a one-dimensional fractional chaos map.
I have the following comments for improvements:
- The abstract is too short and poor, it should include the main finding of this work as well as the motivation behind this work.
- The introduction section is good, however i suggest to add the main contributions of this work as built form points at the last paragraph within the introduction section.
- It seems that no related work section has highlighted this task before. I suggest adding the existing studies that address the encryption algorithm for multiple images.
- Please ensure that all symobls within equations are explained within the text.
- Figure 1 had a poor explanation because it involved 6 subfigures; please explain that clearly within the text.
- The same thing for Fig,2,3,4.
- The presentation of the conducted experiments was worthy, however, it could be explained in a more understandable way, especially in Table 2.
- The conclusion part was too short and poor.
Good Luck
-
Comments on the Quality of English Language
Minor editing of English language required
Round 2
Reviewer 1 Report
Comments and Suggestions for Authors
The author made a lot of revisions to the paper, which is enough to reflect the author's intentions. However, the paper still has the following problems:
The details of key generation should not be omitted from the description. Improvement suggestion: Add an initial key and use the key generation method mentioned in the previous version of the paper to perturb the initial key.
Relevant literature and content on cryptanalysis should be discussed.
The code must be open source and available for review by reviewers and editors on public repositories such as GitHub. At the same time, the open source of the code is beneficial to the spread of the paper and increases the number of citations of the paper.
After the above modifications, it can be considered for acceptance.
Comments on the Quality of English LanguageEnglish needs to be improved by correcting tense and grammatical errors.
Reviewer 2 Report
Comments and Suggestions for Authors
The authors addressed my concerns, however, some grammar checks are needed.
Comments on the Quality of English LanguageSome grammar check is needed.
Round 3
Reviewer 1 Report
Comments and Suggestions for Authors
The author carefully revised the article and responded point-to-point to the questions raised by the reviewers. The amount of revision work is enough to reflect the author's seriousness. I have no more comments.
Comments on the Quality of English LanguageI have no more comments.